# CODECOMPLEX: A TIME-COMPLEXITY DATASET FOR MULTI-LANGUAGE SOURCE CODES

## ABSTRACT

Deciding the computational complexity of algorithms is a really challenging problem, even for human algorithm experts. Theoretically, the problem of deciding the computational complexity of a given program is undecidable due to the famous Halting problem. So, we focus on cases where there are inputs and outputs, and of which we can know if the code is right or wrong. We propose our own dataset CodeComplex, which consists of 4,900 Java codes and 4,900 Python codes submitted to programming competitions by human programmers and their complexity labels annotated by a group of algorithm experts. As far as we are aware, the CodeComplex dataset is by far the largest code dataset for the complexity prediction problem. Then, we present experimental results from several baseline models using the SOTA code understanding neural models such as CodeBERT, GraphCodeBERT, PLBART, CodeT5, CodeT5+ and UniX-coder. We also give an analysis on the difficulties of code complexity and why the models are good/bad on predicting the time complexity. The Code-Complex dataset is available at `https://anonymous.4open.science/r/CodeComplex-Data` and material for reproduction is available at `https://anonymous.4open.science/r/CodeComplex-Models`.

## 1 INTRODUCTION

Worst-case computational (algorithmic) complexity indicates the longest running time $W(n)$ of an algorithm for any input of size $n$. Programmers estimate an upper bound on the time an algorithm needs by analyzing its worst-case time complexity. We often describe the worst-case time complexity using the Big O notation using algebraic expressions. For instance, we denote the time complexity of an algorithm with constant running time, regardless of input size $n$, by $O(1)$. Similarly, we denote the linear-time and quadratic-time algorithms, which require time linear or quadratic in $n$, by $O(n)$ and $O(n^2)$, respectively.

While the worst-case time complexity serves as an effective indicator of algorithm or code efficiency, it is widely recognized that the problem of determining the worst-case time complexity of an algorithm is undecidable (Turing, 1936). Consequently, alternative tractable approaches have been explored to measure the efficiency of an algorithm or a code. These approaches include static code analysis techniques such as cyclomatic complexity (McCabe, 1976), afferent/efferent coupling, and the Master theorem (Bentley et al., 1980). On the other hand, researchers have also delved into dynamic code analysis, which is based on the real-time execution of codes using various test cases to analyze code complexity (Burnim et al., 2009; Hutter et al., 2014; Nogueira, 2012). Dynamic code analysis can detect bugs and measure the execution time and space but requires the generation of suitable test cases and actual code execution with these test cases.

With the rapid progress of programming comprehension models based on large-scale code datasets, the concept of 'AI-powered (AI-assisted) programming' is inching toward reality. OpenAI introduced GitHub Copilot, powered by Codex (Askell et al., 2021), to assist human programmers within integrated development environments (IDEs) by automatically generating codes based on the programming context. More recently, DeepMind introduced AlphaCode (Li et al., 2022) which generates algorithmic programs from natural language descriptions of logical problems encountered in competitive programming. 'AI-powered (AI-assisted) programming' has introduced new people to the programming community. There has been a proliferation of online coding platforms where pro-

grammers log in to write programs using web-based IDE interfaces. These platforms serve various purposes such as collaborative programming, coding interviews, programming competitions, and notably programming education.

We expect that predicting the computational complexity of a code holds significant importance in the education and development of effective programming. However, recent studies (Austin et al., 2021; Jain et al., 2021) have confirmed that research into 'genuine' program understanding AI is still in its early stages. If AI can really understand programs as human programmers do, it is imperative that AI can not only memorize code patterns from training data but also comprehend how to craft 'algorithmically efficient' programs. We can imagine that AI can assist novice programmers in producing efficient codes for algorithmic problems by suggesting code implementations that employ better algorithms compared to the programmers' current codes. Additionally, AI can guide these programmers regarding the relative efficiency of their codes in comparison to average or optimal implementations.

In this paper, we introduce the *CodeComplex* dataset, a novel time complexity benchmark dataset. Our dataset consists of 9,800 program codes (java 4,900, python 4,900) obtained from a competitive programming platform (Codeforces[1]), each of which is annotated with complexity classes by human algorithm experts. As far as we are aware, the CodeComplex dataset is the most extensive public dataset for code complexity prediction, particularly when compared to currently known datasets of its kind, the CoRCoD dataset (Sikka et al., 2020) with 932 Java codes and GFG dataset (Moudgalya et al., 2023) with 1410 C++ and 1373 Python codes.

With CodeComplex, we propose the problem of predicting the worst-case time complexity of a program code. Using cutting-edge deep neural network models and learning algorithms, we provide their baseline performances for code complexity prediction. Our baseline algorithms include the classical machine learning algorithms using hand-crafted features and several state-of-the-art deep learning (DL) algorithms such as CodeBERT, GraphCodeBERT, UnixCoder, PLBART, CodeT5, and CodeT5+.

## 2 CODECOMPLEX: A DATASET OF TIME COMPLEXITY FOR SOURCE CODES

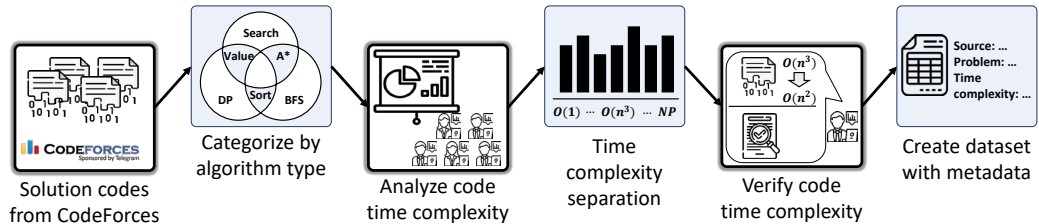

Figure 1: Overall workflow of CodeComplex dataset creation.

CodeComplex is a collection of codes from competitive programming problems written in two languages: Java and Python. Our dataset originates from Codeforces, which sources its data from the CodeContests[2], a competitive programming dataset designed for machine learning applications, developed by DeepMind. It comprises a total of 9,800 codes, with each of the Java and Python languages contributing 4,900 codes. We categorize these codes into seven distinct classes of time complexity: constant ($O(1)$), linear ($O(n)$), quadratic ($O(n^2)$), cubic ($O(n^3)$), logarithmic ($O(\ln n)$, $O(n \ln n)$), and NP-hard. Each complexity class contains a minimum of 500 Java and Python codes. Out of the 9,800 codes, our team annotated the 9,483 codes. The remaining 317 Java codes are from the CoRCoD to prevent duplication, as we have confirmed that they also originate from the CodeContests.

Remark that the CoRCoD (Sikka et al., 2020) is a previous dataset used for the code complexity prediction problem. CoRCoD classifies Java codes into five complexity classes: $O(1)$, $O(n)$, $O(n^2)$, $O(\ln n)$, and $O(n \ln n)$. However, the dataset is problematic in its imbalanced class distribution, as illustrated in Table 1. the dataset size is relatively small for recent DL-based algorithms, consisting of only 929 Java code samples in total. We have expanded the dataset by including two languages,

---

[1]https://codeforces.com/
[2]https://github.com/deepmind/code_contests

Table 1: Statistical difference between CoRCoD and CodeComplex. Numbers in parentheses imply the number of codes from CoRCoD.

| Class | CoRCoD Java | CodeComplex Java | Python |
|---|---|---|---|
| $O(1)$ | 143 | 750 (+ 62) | 791 |
| $O(n)$ | 382 | 779 (+ 117) | 853 |
| $O(n^2)$ | 200 | 765 (+ 48) | 657 |
| $O(n^3)$ | 0 | 601 | 606 |
| $O(\ln n)$ | 54 | 700 (+ 18) | 669 |
| $O(n \ln n)$ | 150 | 700 (+ 72) | 796 |
| NP-hard | 0 | 605 | 528 |
| **Total** | 929 | 4900 (+ 317) | 4900 |

Java and Python, and have considered seven complexity classes instead of five. These complexity classes are among the most commonly encountered in algorithmic problems, and each class has at least 500 codes. This expansion results in a total of 4,900 codes, significantly enhancing the dataset's suitability for research, particularly in the context of recent DL-based models presented in Section 3.1.

## 2.1 ANNOTATION PROCESS

Figure 1 provides an entire process of the CodeComplex creation. Our primary goal in creating CodeComplex is to establish a strong foundation for classifying time complexities accurately. For this reason, we have meticulously designed a procedure to produce a robust dataset with minimal noise and high quality, as outlined below.

Our dataset creation process begins with an examination of algorithmic problems and submitted codes from Codeforces. Since there are diverse nature of submissions, including incorrect codes in various programming languages such as C, C++, Java, and Python, we filter the 'correct' Java and Python codes to form our statistical population. Here, a correct code refers to a code that successfully passes all test cases, including the hidden ones. Next, we categorize the problems based on the problem-solving strategies, leveraging annotations from the CodeContests. Each problem in the dataset is associated with a plausible problem-solving strategy, such as brute force, dynamic programming, greedy, backtracking, and more, as provided in the CodeContests. Finally, we conduct a detailed analysis of each problem, considering the input variables, used data structures, and the workflow of the code. Based on the given variables, we annotate the code of the problem accordingly, aligning it with its specific input characteristics.

**Rules** Our code annotation process comprises the following four key rules:

1. Consider the input size as a parameter to determine the time complexity of a code. We measure the complexity by the largest parameter among them.

2. We take into account the packages and libraries that the code utilizes, such as hashmap, sorting, and string-matching algorithms. The use of these packages and libraries affects the time complexity.

3. When there are multiple test cases within a single input, consider each individual test case as a separate unit for complexity measurement.

4. When a problem provides a fixed constant as input, we classify the case as having a constant time complexity.

It is important to note that algorithmic problems often have various input formats and constraints to ensure solvability in a limited time. These constraints significantly affect the actual running time of a code, deviating from its ideal time complexity. There are scenarios where the problem itself has a quadratic time complexity, but the provided input constraints result in linear running time. Additionally, when faced with multiple input parameters, such as the number of inputs or their

values, we had to decide on which parameter to consider complexity measurement. Furthermore, some code submissions leveraged the parameter constraints described in the problem to pre-compute variables or optimize code execution for improved speed.

## 2.2 Code Augmentation with Consistent Time Complexity

Indeed, while our dataset is currently the largest of its kind, it remains relatively small when compared to the extensive datasets, used in other code-related tasks (Lu et al., 2021; Niu et al., 2023), for training large language models. This size discrepancy raises concerns about overfitting, especially when training these substantial models. In order to mitigate this problem, we employ augmentation techniques to strengthen the dataset's effectiveness during the training process. We approach augmentation with caution, recognizing the potential impact that the augmentation triggers to the original code's time complexity. Our augmentation strategy centers on implementing transformations that modify the structure of a code without altering its semantics. These transformations encompass conversions such as for to while statements, ternary to if statements, and in-lining functions. By adopting these transformations, we aim to solve the problem of size discrepancy without causing variations in code semantics that can affect the code's computational complexity.

## 2.3 Dead Code Elimination

Given that we are working with the codes submitted to a competitive programming platform rather than well-implemented commercial codes, it is common to encounter instances, which contain methods and variables that are not accessed after the declaration. We implement a statistical analysis approach to identify and eliminate these redundant variables and methods.

In the context of Java, which is primarily an object-oriented programming language, a code can be interpreted in a manner that is invariant to permutations. This means that, in a structural aspect of a Java code, the order of class or method declarations does not impact the code semantics. Conversely, Python operates differently. While Python does not support forward declarations in the manner that Java does, it still maintains permutation invariance with regard to function definitions. As long as the functions are defined before any executable code is encountered, the order in which the functions are declared does not alter the code semantics. Thus we are safe to reorder and remove the code segments enclosed by a function, or a class if they are not referred in any part of the code.

Table 4 presents the performance of our model with pre-training objectives on codes where dead codes have been removed. We observe the performance gain of the model trained with pre-training objectives, using the codes after dead code elimination, is marginal on average. However, it is noteworthy that the technique achieves the best performance on longer codes $(1024, \infty]$.

## 3 Experiments

### 3.1 Baseline Models

We benchmark CodeComplex with SOTA models and some traditional ML algorithms as follows.

**ML-based (Sikka et al., 2020)** methods are classic ML classification algorithms such as a decision tree, random forest, and SVM trained with hand-crafted features such as the number of statements, variables, methods, loops, breaks, states, and the existence of data structures (e.g., `HashMap` and `HashSet`) and algorithms (e.g., sorting).

**CodeBERT (Zhong et al., 2020)** is a BERT-like pre-trained language model trained on both natural language (NL) and programming language (PL) like Python, Java, JavaScript.

**GraphCodeBERT (Guo et al., 2021)** is similar to CodeBERT but leverages *data flow* information for pre-training. Note that data flow is a graph that represents a relationship between variables by analyzing where the value of each variable comes from in the entire code.

**UniXCoder (Guo et al., 2022)** utilizes mask attention with prefix adapters to leverage cross-language contents like AST. Encodes AST to a sequence structure that retains all structural information from the tree.

**PLBART (Ahmad et al., 2021)** is a pre-trained model for program understanding and generation that uses both encoder and decoder for pre-training.

**CodeT5 (Yue Wang & Hoi, 2021)**, different from PLBART which only treats codes simply as a sequence of tokens as for NL sentences, relies on code-related features for pre-training such as identifier prediction and tagging.

**CodeT5+ (Wang et al., 2023)** is a flexible encoder-decoder architecture in which component modules can be flexibly combined to suit a wide range of downstream code tasks. Such flexibility is enabled by the proposed mixture of pretraining objectives to mitigate the pretrain-finetune discrepancy.

### 3.2 EXPERIMENTAL SETUP

**Hyperparameters**  For training all Transformer-based models, we use the AdamW optimizer (Loshchilov & Hutter, 2019) along with a learning rate scheduler that includes a warm-up phase with a linear decay. During the fine-tuning pre-trained code understanding models, we set the base learning rate to $2 \cdot 10^{-6}$ and use a weight decay of $1 \cdot 10^{-2}$. We train all models for 15 epochs and utilize a batch size of 6.

**Code Data Preprocessing**  Our initial step involves filtering comments and eliminating `import` and `package` statements that appear on the top of the code using a regular expression, as these statements do not influence the complexity of the code. We utilize javalang parser[3] and Python builtin module ast to transform the code into its AST and extract hierarchical information for each respective language.

Subsequently, for training DL models, we employ the byte-pair encoding (BPE) tokenizer of RoBERTa (Liu et al., 2019) to tokenize the code.

**Dataset Split**  We divide the CodeComplex into training and test datasets using two distinct manners, random split and problem split. The random split, as the name implies, involves randomly allocating the data in a 4:1 ratio for both Java and Python. As a result, the training and test datasets comprise 3,920 and 980 codes, respectively.

In the case of the problem split, we also randomly split the data in a similar ratio but ensure that the training and test datasets do not share any common problems. In other words, instead of randomly selecting codes, we randomly select problems and assign the entire codes associated with the chosen problems to the test dataset until it constitutes approximately 1/5 of the total 4,900 codes. This approach is based on the premise that codes solving the same problem tend to exhibit similar code structures and give a false attribute. When employing the problem split, we conduct five-fold validation and calculate the average accuracy from these iterations. This is because the prediction accuracy is often highly sensitive to the choice of problems.

### 3.3 EXPERIMENTAL RESULTS

Table 2 shows the experimental results of the baselines. The results show that splitting the dataset with different strategies significantly affects the prediction performance. While the classical ML methods based on the hand-crafted features show the worst performance on both random split and problem split, pre-trained models for program languages show a greater performance difference on random split and problem split exhibit much better performance on the random split. This implies that the models learn the similarity of codes for the same problem instead of the operational semantics of codes for complexity prediction.

**ML vs DL for Complexity Prediction**  It can be clearly seen that ML methods perform poorly on both splits. It should be noted that the complexity prediction accuracy is much lower in our result than the result reported by Sikka et al. (2020). The first reason is that the CodeComplex dataset has more classes (7) than the CoRCoD dataset (5). Second, we speculate that the training and test sets have a soft overlap in the experiments as we confirm that there were duplicated codes in the

---

[3]https://github.com/c2nes/javalang

Table 2: Complexity prediction performance on different dataset splits.

| Method | Random | | Problem | |
|---|---|---|---|---|
| | Java | Python | Java | Python |
| Decision Tree | 50.6% | 48.5% | 46.9% | 43.7% |
| Random Forest | 54.4% | 50.3% | 45.3% | 45.5% |
| SVM | 38.1% | 33.2% | 24.7% | 20.4% |
| CodeBERT | 73.5% | 70.4% | 61.4% | 52.0% |
| GraphCodeBERT | 82.4% | 77.1% | 57.2% | 57.8% |
| UniXCoder | 82.5% | 81.3% | 57.9% | 55.8% |
| PLBART | 86.3% | 74.4% | 63.8% | 55.4% |
| CodeT5 | 79.3% | 65.9% | 60.2% | 49.8% |
| CodeT5+ | 83.3% | 75.4% | 57.4% | 50.0% |

Table 3: Complexity prediction accuracy of classification methods for each complexity class.

| Category | Method | $O(1)$ | $O(n)$ | $O(n^2)$ | $O(n^3)$ | $O(\ln n)$ | $O(n \ln n)$ | NP-hard |
|---|---|---|---|---|---|---|---|---|
| ML-based | Decision Tree | 71.8% | 15.4% | 76.1% | 30.5% | 50.5% | 58.8% | 22.2% |
| | Random Forest | 68.1% | 18.0% | 38.4% | 25.0% | 48.6% | 68.0% | 67.9% |
| | SVM | 42.6% | 17.6% | 13.1% | 6.0% | 27.1% | 24.9% | 77.0% |
| Encoder (Java) | CodeBERT | 85.5% | 60.0% | 20.0% | 33.3% | 69.1% | 71.7% | 83.9% |
| | GraphCodeBERT | 97.4% | 53.3% | 32.7% | 34.0% | 52.8% | 77.7% | 84.9% |
| | UniXCoder | 58.7% | 53.67 | 8.96 | 33.2 | 67.4 | 67.7 | 55.6 |
| Encoder (Python) | CodeBERT | 66.9% | 64.4% | 44.5% | 34.9% | 41.6% | 64.8% | 26.9% |
| | GraphCodeBERT | 67.0% | 56.5% | 50.0% | 50.2% | 64.9% | 67.8% | 33.7% |
| | UniXCoder | 58.7% | 62.2% | 48.3 | 44.0 | 64.9 | 64.1 | 38.4 |
| Enc-Dec (Java) | PLBART | 86.4% | 53.3% | 37.8% | 33.5% | 58.5% | 76.5% | 87.5% |
| | CodeT5 | 80.5% | 42.7% | 43.4% | 29.4% | 65.8% | 72.8% | 85.1% |
| | CodeT5+ | 89.7% | 49.3% | 15.5% | 24.2% | 64.8% | 75.5% | 86.4% |
| Enc-Dec (Python) | PLBART | 70.1% | 63.9% | 47.8% | 40.6% | 59.5% | 61.3% | 25.9% |
| | CodeT5 | 65.0% | 44.5% | 39.0% | 37.1% | 57.3% | 58.5% | 40.3% |
| | CodeT5+ | 62.2% | 55.6% | 24.7% | 29.6% | 70.3% | 65.5% | 27.1% |

929 codes of the CoRCoD dataset. We find that 50 codes from 929 codes have exactly equivalent codes and two codes have 'almost equivalent' codes (which become exactly equivalent after filtering comments) within the dataset.

While ML methods do not perform well even on a random split, DL models perform robustly on a random split (around 80% accuracy). Due to a certain amount of problem overlap in random split, DL models make use of specific token names consistently used for the same problem and structural similarities for predicting the same complexity class with the code seen during the training. The performance boost compared to the case of problem split comes from the fact that DL models successfully exploit the token names from raw codes.

**Model structure effectiveness on code understanding**  Looking at the results of table 3 shows us that DL models we have invested in have trouble in classifying the cubic time complexity ($O(n^3)$). In our experiments, the cubic time complexity is predicted as the quadratic time complexity ($O(n^2)$) often. We tend to think this is due to the code flow where the quadratic part is outlined to a function, and the main function only consists of a linear loop. The models seem to fail at linking the function name to its declaration. PLBART seems to be better in linking usage to definition since PLBART is pre-trained to understand program syntax and logical flow.

Table 4: Prediction performance on different code lengths

| Method | (0, 256] | (256, 512] | (512, 1024] | (1024, $\infty$] |
|---|---|---|---|---|
| Decision Tree | 57.2% | 45.6% | 40.0% | 38.2% |
| Random Forest | 62.3% | 46.8% | 40.6% | 26.4% |
| SVM | 48.9% | 18.1% | 18.1% | 16.6% |
| CodeBERT(Java) | 72.4% | 62.8% | 60.7% | 48.0% |
| CodeBERT(Python) | 56.9% | 46.9% | 37.5% | 22.8% |
| GraphCodeBERT(Java) | 74.6% | 61.7% | 49.8% | 39.4% |
| GraphCodeBERT(Python) | 60.3% | 57.8% | 44.1% | 30.8% |
| UniXCoder(Java) | 58.6% | 54.4% | 43.2% | 31.2% |
| UniXCoder(Python) | 58.6% | 54.4% | 43.2% | 31.2% |
| PLBART(Java) | 74.3% | 65.1% | 62.5% | 52.8% |
| PLBART(Python) | 60.6% | 49.4% | 39.9% | 23.2% |
| CodeT5(Java) | 69.5% | 56.5% | 52.4% | 42.4% |
| CodeT5(Python) | 53.6% | 48.1% | 36.5% | 19.5% |
| CodeT5+(Java) | 72.8% | 63.5% | 53.0% | 44.4% |
| CodeT5+(Python) | 56.4% | 42.4% | 30.7% | 29.8% |

**Relationship between Code Length and Accuracy**    Table 4 shows the prediction performance of models on codes of different lengths. We partition the codes in the test set into four groups according to the length of sequences processed by the javalang and pythons built-in tokenizer and calculate the prediction accuracy for each group. We can confirm the clear tendency that the prediction performance degrades as the length of code increases in every model. In fact, after applying dead code elimination(unreachable or unused codes), results were improved across every model. Also, models like CodeT5 have been pre-trained to a token length of 256 which makes the situation worse. As we have argued before, linking functions to their definitions seems to be crucial, and longer code leads to increasing difficulty.

**Error Case Analysis**    Table 3 and Fig. 2 show the type of errors model more frequently makes. We can see that the model is easily confused on polynomial-time algorithms including $O(n), O(n^2)$ and $O(n^3)$. Our model makes many mispredictions for $O(n)$ codes by predicting other classes uniformly except $O(2^n)$ and is also frequently confused between quadratic and cubic algorithms.

Fig. 3 shows two failure cases where our model fails to predict the correct complexity class of codes. Fig. 3a is the case where our model predicts quadratic time complexity for the code with linear time complexity. At a glance, the code actually seems to be in $O(n^2)$ due to the nested `for` loops. However, the number of iterations is actually controlled by an integer variable `k`. Another example in Fig. 3b is also interesting. Our model predicts the complexity class as $O(n^2)$ but the actual complexity is $O(n^3)$ because the method inside the nested `for` loops named `lowestCost` runs in linear time in the size of the input. From these error cases, we can deduce that our model focuses on the computational structure of a code rather than merely the token distribution of a code.

## 4    RELATED WORK

**Analyzing Time Complexity of Programs**    McCabe (1976) introduced a metric for quantitatively measuring program complexity known as *cyclomatic complexity*. In essence, cyclomatic complexity quantifies the number of linearly dependent paths in a program. Bentley et al. (1980) introduced the *Master theorem*, which is a valuable tool for analyzing the time complexity of divide-and-conquer algorithms. Its analysis is based on expressing the algorithm's time complexity as a recurrence relation and providing methods to solve this relation.

In a more recent study, Sikka et al. (2020) focused on machine learning-based methods for code complexity prediction. They released a novel code dataset containing 929 Java codes, each annotated with runtime complexities and proposed baselines of machine learning-based models with

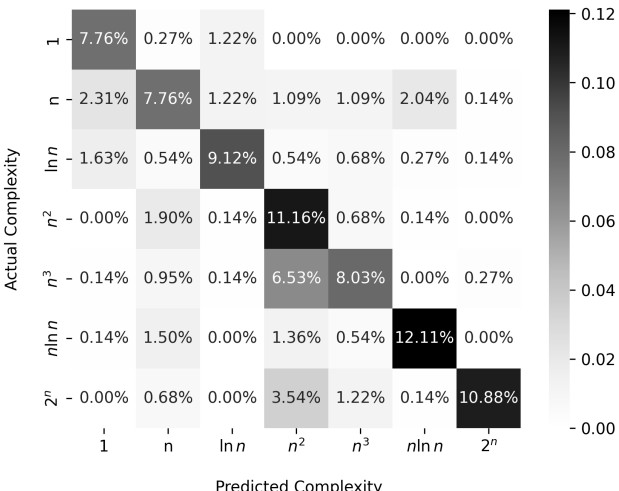

Figure 2: Confusion matrices for the predictions on problem split with CodeBERT

hand-engineered features. Their approach involved extracting various features from the code such as counts of loops, methods, variables, jumps, breaks, switches and the presence of specific data structures or algorithms such as priority queue, hash map, hash set, and sorting functions. Subsequently, they applied machine-learning classification algorithms, including $K$-means, random forest, decision tree, SVM and more, to make predictions based on these features. Additionally, they also reported similar performance results by embedding the graph structure of a program's AST with a neural graph embedding framework, graph2vec (Narayanan et al., 2017). Prenner & Robbes (2021) analyzed the performance of the pre-trained programming language understanding model such as CodeBERT (Feng et al., 2020) for predicting code complexity. They presented the experimental results that the pre-trained model can be a promising solution to the problem.

Most recently, Moudgalya et al. (2023) approached the question of analyzing time and space complexity by the utilization of language models. They used codes scraped from GeeksForGeeks and CoRCoD alongside our preceding dataset which consists of 3,803 Java codes. They showed that pre-trained language models such as GraphCodeBERT can be fine-tuned to predict time and space complexity using such datasets.

**Programming Language Understanding Models** There have been numerous studies on pre-training methods for understanding programming languages. Feng et al. (2020) proposed Code-BERT, which is a RoBERTa-based model pre-trained on multiple programming languages with masked language modeling (MLM) and replaced token prediction (RTD) objectives. Guo et al. (2021) introduced GraphCodeBERT which is strengthened from CodeBERT by incorporating data flow information in the pre-training stage. Jiang et al. (2021) introduced TreeBERT, a tree-based pre-trained model that focuses on utilizing the extracted tree structure by encoding an AST as a set of composition paths. TreeBERT is trained by two novel objectives called tree-masked language modeling (TMLM) and node order prediction (NOP). Rozière et al. (2021) investigated another programming language-oriented pre-training objective called DOBF, which is based on deobfuscation of identifier names in source code. Note that we do not use TreeBERT and DOBF as our baseline as they are mainly for code generation tasks not for the classification task.

Recently, Ahmad et al. (2021) proposed PLBART (Program and Language BART), which learns the interaction between program codes and natural language descriptions by leveraging the idea of denoising autoencoder that uses a bidirectional encoder and an auto-regressing decoder. Yue Wang & Hoi (2021) introduced CodeT5, which leverages the code-specific characteristics in the pre-training stage by employing the new objectives such as masked random token prediction, masked identifier prediction, and identifier prediction objectives.

```
1   int n = nextInt();
2   int k = nextInt();
3   int[] a = new int[n];
4   for (int i = 0; i < n; i++) {
5       a[i] = nextInt();
6   }
7   Set<Integer> set = new
        HashSet<Integer>();
8   for (int i = 0; i < a.length;
        i++) {
9       set.add(a[i]);
10      if (set.size() == k) {
11          Set<Integer> set2 = new
                HashSet<Integer>();
12          for (int j = i; j >= 0;
                j--) {
13              set2.add(a[j]);
14              if (set2.size() ==
                    k) {
15                  out.print((j + 1)
                        + " " + (i +
                        1));
16                  out.close();
17                  return;
18              }
19          }
20      }
21  }
```

```
1   public static void
        main(String[] args){
2       for (int i = 0; i < n;
            i++) {
3           for (int j = 0; j < m;
                j++) {
4               if (steps % 2 != 0) {
5                   out.print(-1 + "
                        ");
6               } else {
7                   out.print(2 *
                        lowestCost(i,
                        j, steps / 2)
                        + " ");
8               }
9           }
10      }
11  }
12  private long lowestCost(int
        i, int j, int distance) {
13      long minDist =
            Long.MAX_VALUE;
14      if (i > 0)
15          minDist =
                Math.min(minDist,
                distI[i - 1][j] +
                lowestCost(i - 1,
                j, distance - 1));
16      ...
17      return minDist;
18  }
```

(a) A code snippet from a program whose complexity is predicted as $O(n^2)$ by our model while the actual complexity is in $O(n)$.

(b) A code snippet from a program whose complexity is predicted as $O(n^2)$ by our model while the actual complexity is in $O(n^3)$.

Figure 3: Failure examples of the most frequent mispredictions discovered from a confusion matrix.

## 5 CONCLUSION

We present CodeComplex, a multi-language benchmark dataset for classifying code running time. We showcase the usage of pre-trained SOTA models and classical algorithms for predicting the computational complexity with CodeComplex. The results show that improvements could be made, and that AI can indeed be a future companion on coding. CodeComplex is the biggest dataset of its kind, and we used precautious steps in producing the dataset. We hope CodeComplex to bring a spark in this field. For future work

## REPRODUCIBILITY

For reproducing the results of our paper, please refer to the models and data link in the abstract. The material contains data and models along with data preprocessing and the splitting method by problem into five-fold splits for verifying the generalization of our model to unseen problems.

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

## A    OVERVIEW ON CODECOMPLEX DATASET

Our dataset construction process owes much to the recently released dataset called the CodeContests[4], a competitive programming dataset for machine learning by DeepMind. We constructed a dataset with the codes from the CodeContests dataset that are again sourced from the coding competition platform Codeforces. Our dataset contains 4,120 codes in seven complexity classes, where

---

[4]https://github.com/deepmind/code_contests

there are new 500 Java source codes annotated with each complexity class. The seven complexity classes are constant ($O(1)$), linear ($O(n)$), quadratic ($O(n^2)$), cubic ($O(n^3)$), $O(\ln n)$, $O(n \ln n)$, and NP-hard. We also re-use 317 Java codes from CoRCoD as we confirmed that they also belong to the CodeContests dataset as the other 3803 codes during the dataset creation process.

For constructing the dataset, we asked twelve human annotators who have more than five years of programming experience and algorithmic expertise to inspect the codes manually and classify them into one of the seven complexity classes. Once each human annotator reported the initial result, we collected the annotation results and inspected them once again by assigning the initial result to two different annotators other than the initial annotator. Finally, we have collected 3803 complexity annotated codes in which there are 500 codes for each complexity class.

First, we selected several problems that are expected to belong to one of the considered complexity classes and submitted codes for the problems from Codeforces. The submitted codes contain both correct and incorrect solutions, and they are implemented in various programming languages such as C, C++, Java, and Python. We sorted out only the correct Java codes for our dataset construction.

In the second step, before delving into the time complexity of problems, we divide the problems by the problem-solving strategy such as sorting, DP (dynamic programming), divide-and-conquer, DFS (depth-first search), BFS (breadth-first search), A*, and so on. This is because it is helpful to know the type of problem-solving strategy used to solve the problem for human annotators to analyze the time complexity, and problems solved by the same strategy tend to have similar time complexity.

Third, we uniformly assign problems and correct codes for the problems to human annotators and let them carefully examine the problem-code pairs to label the time complexity of the codes. Notice that there can be solutions with different time complexities for a problem depending on how to actually implement the solutions. We, therefore, provide a specific guideline that contains instructions and precautions to annotators so that human annotators can assign correct and consistent labels to the assigned codes.

After the initial annotation process, we collect the results and assign them to different annotators to carefully cross-check the correctness of the initial annotation results. Primarily, we instruct the annotators again to carefully verify the results in accordance with the precautions provided in the annotation guideline.

## A.1 Further Details on CodeComplex Dataset Construction

We gathered 128,000,000 submissions of Codeforces, where 4,086,507 codes are implemented in Java language. After discarding the incorrect codes (that do not pass all the test cases), there are 2,034,925 codes and 7,843 problems. Then the problems are split with their tags (e.g. sorting, dfs, greedy, etc) and given to the annotators with the guidelines in Section A.2. We were able to gather around 500 problems and 15,000 codes for the seven complexity classes.

As the complexity of codes for the same problem can vary depending on the implemented algorithms, it is obvious that the codes we inspect also have various complexity classes. However, we only target seven complexity classes that are the most frequently used complexity classes for algorithmic problems. Accordingly, there were some codes we inspected which belong to other complexity classes such as $O(n^5)$ or $O(\ln \ln n)$. We inspected around 800 problems and found out that the complexity classes of approximately 15% of the problems belong outside the chosen complexity classes. Although it is still possible that one might implement codes with complexity class that falls into the seven complexity classes, we simply rule out the problems from our dataset to ease the annotation process.

During this process, we found out that many codes are not optimal for the given problem and some codes are too difficult to analyze due to their complex code structure. Moreover, there are many codes with a number of methods that are never used, mainly because the codes come from a coding competition platform and participants prefer just to include the methods that are frequently used in problem-solving regardless of the actual usage of the methods.

Our dataset, the CodeComplex dataset is constructed from the code instances from the Codeforces platform which are recently revealed from AlphaCode. As Codeforces is a coding competition platform, our dataset consists of codes that implement various algorithms that are designed to solve

algorithmically challenging problems. Our dataset offers a larger number of source codes with a broader category of complexity classes compared to the sole existing complexity prediction benchmark dataset, CoRCoD dataset (Sikka et al., 2020). In the section below, we share the detailed guidelines provided to human annotators for the precise code complexity annotation process.

## A.2 GUIDELINE OF PRODUCTION

1. Check the variables that are described in the algorithm problems. Each algorithm implementation can have many variable instances and we only consider the variables that are given as inputs from the problems for calculating the time complexity.

   * For convenience, we use $n$ and $m$ in the guideline to denote the input variable and $|n|$ and $|m|$ to denote the size of $n$ and $m$.

2. Considering the input variable from the prior step, follow the below instructions for each input type and calculate the time complexity.

   (a) When only a number $n$ is given as an input, calculate the time complexity proportional to $n$. Do the same thing when there are two or more variables. For instance, when only $n$ is given as an input, the variable used to denote the time complexity of a code is $n$.

   (b) When a number $n$ and $m$ numeric instances are given as inputs, calculate the time complexity proportional to the one with higher complexity. For instance, when $m = n^2$, we compute the complexity of a code with $m$. If the implemented algorithm runs in $O(n^2) = O(m)$, it belongs to the linear complexity class.

   (c) If the input is given as constant values, the complexity of a given code also belongs to the constant class. For instance, if an algorithm problem states that exactly 3 numeric values are given as inputs, the solution code only uses the constant number of operations. Therefore, the code belongs to the constant class.

3. Consider the case where the code utilizes the input constraints of the problem. When the input is given by $n \leq a$, the code can use the fixed value $a$ in the problem instead of using $n$. Mark these codes as unsuitable.

4. Consider the built-in library that the implemented algorithm is using (e.g. HashMap, sort, etc.) to calculate the time complexity of an entire code. For instance, given $n$ numeric instances as inputs, when an implemented algorithm uses $O(n)$ iterations of built-in sort algorithm for $n$ numeric instances, the time complexity for the algorithm is $O(n^2 \ln n)$.

5. When the code has unreachable codes, only consider the reachable code.

6. Mark the item that does not belong to any of the 7 complexity classes.

## B   FAILURE CASES

- Listing 1 exhibits a failure example where our model predicts $O(2^n)$ for a code with $O(\ln n)$ complexity. We suspect that the primary reason is the usage of bitwise operators. When we filter the codes that use any bitwise operator at least once from our CodeComplex dataset, about 56% of the codes belong to the class $O(2^n)$, which implies NP-hardness. We find that many implementations for NP-hard problems rely on bitwise operators as they can efficiently manage the backtracking process by manipulating bit-level flags.

Listing 1: A failure example of our model (GT: $O(\ln n)$, Prediction: $O(2^n)$).

```
1   public class mad {
2     public static void main(String[] args) {
3         Scanner sc = new Scanner(System.in);
4         int cura = 0, curb = 0;
5         int ver;
6         System.out.println("? 0 0");
7         System.out.flush();
8         ver = sc.nextInt();
9         for (int i = 29; i >= 0; i--) {
10            System.out.println("? " + (cura + (1 << i)) + " " + curb);
```

```
11          System.out.flush();
12          int temp1 = sc.nextInt();
13          System.out.println("? " + cura + " " + (curb + (1 << i)));
14          System.out.flush();
15          int temp2 = sc.nextInt();
16          if (temp1 != temp2) {
17              if (temp2 == 1) {
18                  cura += (1 << i);
19                  curb += (1 << i);
20              }
21          } else {
22              if (ver == 1) cura += (1 << i);
23              if (ver == -1) curb += (1 << i);
24              ver = temp1;
25          }
26      }
27      System.out.println("! " + cura + " " + curb);
28  }
29 }
```

- Listing 2 demonstrates the case when our model predicts constant time complexity $O(1)$ for a code that runs in $O(n)$ time. We suspect that our model may have ignored the existence of the check method which actually determines the $O(n)$ time complexity or considered the argument of check as constant.

Listing 2: A failure example of our model (GT: $O(n)$, Prediction: $O(1)$).

```
1  public class abc {
2    public static int check(StringBuilder s) {
3        int countRemove = 0;
4        if (!s.toString().contains("xxx")) return countRemove;
5        else {
6            for (int i = 1; i < s.length() - 1; i++) {
7                if (s.charAt(i - 1) == 'x' && s.charAt(i) == 'x' &&
                      s.charAt(i + 1) == 'x') {
8                    countRemove++;
9                }
10           }
11           return countRemove;
12       }
13   }
14
15   public static void main(String[] args) {
16       Scanner sc = new Scanner(System.in);
17       int n = sc.nextInt();
18       String s = sc.next();
19       StringBuilder sb = new StringBuilder("");
20       sb.append(s);
21       System.out.println(check(sb));
22   }
23 }
```

- Listing 3 shows the case where our model predicts the quadratic time complexity $O(n^2)$ when the ground-truth label is $O(n \ln n)$. We guess that our model simply translates the nested for loops into the quadratic time complexity. However, the outer loop is to repeat each test case and therefore should be ignored. Then, the $O(n \ln n)$ complexity can be determined by the sort function used right before the nested loops.

Listing 3: A failure example of our model (GT: $O(n \ln n)$, Prediction: $O(n^2)$).

```
1  ppublic class round111A {
```

```java
2      public static void main(String[] args) {
3          Scanner sc = new Scanner(System.in);
4          int n = sc.nextInt();
5          int[] coins = new int[n];
6          for (int i = 0; i < n; ++i) coins[i] = sc.nextInt();
7          Arrays.sort(coins);
8          int ans = (int) 1e9;
9          for (int i = 1; i <= n; ++i) {
10             int sum1 = 0;
11             int c = 0;
12             int j = n - 1;
13             for (j = n - 1; j >= 0 && c < i; --j, ++c) {
14                 sum1 += coins[j];
15             }
16             int sum2 = 0;
17             for (int k = 0; k <= j; ++k) sum2 += coins[k];
18             if (sum1 > sum2) {
19                 System.out.println(i);
20                 return;
21             }
22         }
23     }
24  }
```

- Listing 4 shows the case when our model is confused exponential complexity $O(2^n)$ with quadratic complexity $O(n^2)$. The code actually runs in exponential time in the worst-case but our model simply returns quadratic time complexity as it does not take into account the recursive nature of the method `solve`.

Listing 4: A failure example of our model (GT: $O(2^n)$, Prediction: $O(n^2)$).

```java
1  public class D {
2      static int n, KA, A;
3      static int[] b;
4      static int[] l;
5      static double ans = 0;
6
7      public static void main(String[] args) throws IOException {
8          Scanner in = new Scanner(System.in);
9          n = in.nextInt();
10         KA = in.nextInt();
11         A = in.nextInt();
12         b = new int[n];
13         l = new int[n];
14         for (int i = 0; i < l.length; i++) {
15             b[i] = in.nextInt();
16             l[i] = in.nextInt();
17         }
18         dp = new double[n + 2][n + 2][n * 9999 + 2];
19         go(0, KA);
20         System.out.printf("%.6f\n", ans);
21     }
22
23     public static void go(int at, int k) {
24         if (at == n) {
25             ans = Math.max(ans, solve(0, 0, 0));
26             return;
27         }
28         for (int i = 0; i <= k; i++) {
29             if (l[at] + i * 10 <= 100) {
30                 l[at] += i * 10;
31                 go(at + 1, k - i);
32                 l[at] -= i * 10;
```

```
33                }
34            }
35        }
36
37        static double dp[][][];
38
39        public static double solve(int at, int ok, int B) {
40            if (at == n) {
41                if (ok > n / 2) {
42                    return 1;
43                } else {
44                    return (A * 1.0) / (A * 1.0 + B);
45                }
46            }
47            double ret = ((l[at]) / 100.0) * solve(at + 1, ok + 1, B) +
                    (1.0 - ((l[at]) / 100.0)) * solve(at + 1, ok, B + b[at]);
48            return ret;
49        }
50
51    }
```

- Listing 5 shows the case when our model predicts $O(\ln n)$ for a code with $O(n^2)$ complexity. It is easily seen that the `inversions` function determines the quadratic time complexity by the nested `for` loops. We suspect that somehow our model does not take into account the `inversions` function in the complexity prediction and instead focuses on the modulo (%) operator to draw the wrong conclusion that the complexity is in $O(\ln n)$.

Listing 5: A failure example of our model (GT: $O(n^2)$, Prediction: $O(\ln n)$).

```java
1  public class maestro {
2      public static long inversions(long[] arr) {
3          long x = 0;
4          int n = arr.length;
5          for (int i = n - 2; i >= 0; i--) {
6              for (int j = i + 1; j < n; j++) {
7                  if (arr[i] > arr[j]) {
8                      x++;
9                  }
10             }
11         }
12         return x;
13     }
14
15     public static void main(String[] args) {
16         Scanner sc = new Scanner(System.in);
17         int n = sc.nextInt();
18         long[] arr = new long[n];
19         for (int i = 0; i < n; i++) arr[i] = sc.nextLong();
20         long m = sc.nextLong();
21         long x = inversions(arr) % 2;
22         for (int i = 0; i < m; i++) {
23             int l = sc.nextInt() - 1;
24             int r = sc.nextInt() - 1;
25             if ((r - l + 1) % 4 > 1) x = (x + 1) % 2;
26             if (x == 1) System.out.println("odd");
27             else System.out.println("even");
28         }
29     }
30 }
```

## C    FURTHER DETAILS ON DEAD CODE ELIMINATION

In a broad sense, the dead code includes redundant code, unreachable code, oxbow code, and so on. We only focus on eliminating unreachable codes, mainly methods and classes that are declared but used nowhere in the code. In order to find such dead codes, we first parse a Java code into an AST and discover methods and classes that do not exist in any method call, class declaration, and arguments of methods. Once we discover such unused methods and classes, we remove the codes corresponding to the declarations of these methods and classes.

Listings 6 and 7 show a running example of the dead code elimination process. From the code in Listing 6, we can obtain the code in Listing 7 by applying the dead code elimination. It is readily seen that the number of lines decreases from 211 to 101 by the elimination process. In fact, our model predicts $O(\ln n)$ and $O(1)$ for the complexity of the code before and after dead code elimination, respectively, while the actual complexity of the code is $O(1)$.

Listing 6: An example code containing many dead codes such as unused methods and variables.

```
1   public class Main {
2       static long mod = ((long) 1e9) + 7;
3
4       public static int gcd(int a, int b) {
5           if (b == 0) return a;
6           else return gcd(b, a % b);
7       }
8
9       public static long pow_mod(long x, long y) {
10          long res = 1;
11          x = x % mod;
12          while (y > 0) {
13              if ((y & 1) == 1) res = (res * x) % mod;
14              y = y >> 1;
15              x = (x * x) % mod;
16          }
17          return res;
18      }
19
20      public static int lower_bound(int[] arr, int val) {
21          int lo = 0;
22          int hi = arr.length - 1;
23          while (lo < hi) {
24              int mid = lo + ((hi - lo + 1) / 2);
25              if (arr[mid] == val) {
26                  return mid;
27              } else if (arr[mid] > val) {
28                  hi = mid - 1;
29              } else lo = mid;
30          }
31          if (arr[lo] <= val) return lo;
32          else return -1;
33      }
34
35      public static int upper_bound(int[] arr, int val) {
36          int lo = 0;
37          int hi = arr.length - 1;
38          while (lo < hi) {
39              int mid = lo + ((hi - lo) / 2);
40              if (arr[mid] == val) {
41                  return mid;
42              } else if (arr[mid] > val) {
43                  hi = mid;
44                  ;
45              } else lo = mid + 1;
46          }
47          if (arr[lo] >= val) return lo;
```

```java
48          else return -1;
49      }
50
51      public static void main(String[] args) throws java.lang.Exception {
52          Reader sn = new Reader();
53          Print p = new Print();
54          int n = sn.nextInt();
55          while ((n--) > 0) {
56              int a = sn.nextInt();
57              int b = sn.nextInt();
58              int small = Math.min(a, b);
59              int large = Math.max(a, b);
60              long steps = 0;
61              while (small != 0) {
62                  steps += (long) (large / small);
63                  int large1 = small;
64                  small = large % small;
65                  large = large1;
66              }
67              p.printLine(Long.toString(steps));
68          }
69          p.close();
70      }
71  }
72
73  class Pair implements Comparable<Pair> {
74      int val;
75      int in;
76
77      Pair(int a, int b) {
78          val = a;
79          in = b;
80      }
81
82      @Override
83      public int compareTo(Pair o) {
84          if (val == o.val) return Integer.compare(in, o.in);
85          else return Integer.compare(val, o.val);
86      }
87  }
88
89  class Reader {
90      final private int BUFFER_SIZE = 1 << 16;
91      private DataInputStream din;
92      private byte[] buffer;
93      private int bufferPointer, bytesRead;
94
95      public boolean isSpaceChar(int c) {
96          return c == ' ' || c == '\n' || c == '\r' || c == '\t' || c ==
                  -1;
97      }
98
99      public Reader() {
100         din = new DataInputStream(System.in);
101         buffer = new byte[BUFFER_SIZE];
102         bufferPointer = bytesRead = 0;
103     }
104
105     public Reader(String file_name) throws IOException {
106         din = new DataInputStream(new FileInputStream(file_name));
107         buffer = new byte[BUFFER_SIZE];
108         bufferPointer = bytesRead = 0;
109     }
110
111     public String readLine() throws IOException {
```

```
112        byte[] buf = new byte[64];
113        int cnt = 0, c;
114        while ((c = read()) != -1) {
115            if (c == '\n') break;
116            buf[cnt++] = (byte) c;
117        }
118        return new String(buf, 0, cnt);
119    }
120
121    public String readWord() throws IOException {
122        int c = read();
123        while (isSpaceChar(c)) c = read();
124        StringBuilder res = new StringBuilder();
125        do {
126            res.appendCodePoint(c);
127            c = read();
128        } while (!isSpaceChar(c));
129        return res.toString();
130    }
131
132    public int nextInt() throws IOException {
133        int ret = 0;
134        byte c = read();
135        while (c <= ' ') c = read();
136        boolean neg = (c == '-');
137        if (neg) c = read();
138        do {
139            ret = ret * 10 + c - '0';
140        } while ((c = read()) >= '0' && c <= '9');
141        if (neg) return -ret;
142        return ret;
143    }
144
145    public long nextLong() throws IOException {
146        long ret = 0;
147        byte c = read();
148        while (c <= ' ') c = read();
149        boolean neg = (c == '-');
150        if (neg) c = read();
151        do {
152            ret = ret * 10 + c - '0';
153        } while ((c = read()) >= '0' && c <= '9');
154        if (neg) return -ret;
155        return ret;
156    }
157
158    public double nextDouble() throws IOException {
159        double ret = 0, div = 1;
160        byte c = read();
161        while (c <= ' ') c = read();
162        boolean neg = (c == '-');
163        if (neg) c = read();
164        do {
165            ret = ret * 10 + c - '0';
166        } while ((c = read()) >= '0' && c <= '9');
167        if (c == '.') {
168            while ((c = read()) >= '0' && c <= '9') {
169                ret += (c - '0') / (div *= 10);
170            }
171        }
172        if (neg) return -ret;
173        return ret;
174    }
175
176    private void fillBuffer() throws IOException {
```

```
177         bytesRead = din.read(buffer, bufferPointer = 0, BUFFER_SIZE);
178         if (bytesRead == -1) buffer[0] = -1;
179     }
180
181     private byte read() throws IOException {
182         if (bufferPointer == bytesRead) fillBuffer();
183         return buffer[bufferPointer++];
184     }
185
186     public void close() throws IOException {
187         if (din == null) return;
188         din.close();
189     }
190 }
191
192 class Print {
193     private final BufferedWriter bw;
194
195     public Print() {
196         bw = new BufferedWriter(new OutputStreamWriter(System.out));
197     }
198
199     public void print(String str) throws IOException {
200         bw.append(str);
201     }
202
203     public void printLine(String str) throws IOException {
204         print(str);
205         bw.append("\n");
206     }
207
208     public void close() throws IOException {
209         bw.close();
210     }
211 }
```

Listing 7: A code obtained from Listing 7 by eliminating dead codes.

```
1  public class Main {
2      static long mod = ((long) 1e9 + 7);
3
4      public static int gcd(int a, int b) {
5          if ((b == 0)) return a;
6          else return gcd(b, (a % b));
7      }
8
9      public static void main(String[] args) throws java.lang.Exception {
10         Reader sn = new Reader();
11         Print p = new Print();
12         int n = sn.nextInt();
13         while ((n > 0)) {
14             int a = sn.nextInt();
15             int b = sn.nextInt();
16             int small = Math.min(a, b);
17             int large = Math.max(a, b);
18             long steps = 0;
19             while ((small != 0)) {
20                 steps += (long) (large / small);
21                 int large1 = small;
22                 small = (large % small);
23                 large = large1;
24             }
25             p.printLine(Long.toString(steps));
26         }
```

```
27          p.close();
28      }
29  }
30
31  class Reader {
32      final private int BUFFER_SIZE = (1 << 16);
33      private DataInputStream din;
34      private byte[] buffer;
35      private int bufferPointer, bytesRead;
36
37      public boolean isSpaceChar(int c) {
38          return ((((c == ' ') || (c == '\n')) || (c == '\r')) || (c ==
                '\t')) || (c == -1));
39      }
40
41      public Reader() {
42          din = new DataInputStream(System.in);
43          buffer = new byte[BUFFER_SIZE];
44          bufferPointer = bytesRead = 0;
45      }
46
47      public Reader(String file_name) throws IOException {
48          din = new DataInputStream(new FileInputStream(file_name));
49          buffer = new byte[BUFFER_SIZE];
50          bufferPointer = bytesRead = 0;
51      }
52
53      public int nextInt() throws IOException {
54          int ret = 0;
55          byte c = read();
56          while ((c <= ' ')) c = read();
57          boolean neg = (c == '-');
58          if (neg) c = read();
59          do {
60              ret = (((ret * 10) + c) - '0');
61          } while ((((c = read()) >= '0') && (c <= '9')));
62          if (neg) return -ret;
63          return ret;
64      }
65
66      private void fillBuffer() throws IOException {
67          bytesRead = din.read(buffer, bufferPointer = 0, BUFFER_SIZE);
68          if ((bytesRead == -1)) buffer[0] = -1;
69      }
70
71      private byte read() throws IOException {
72          if ((bufferPointer == bytesRead)) fillBuffer();
73          return buffer[bufferPointer++];
74      }
75
76      public void close() throws IOException {
77          if ((din == null)) return;
78          din.close();
79      }
80  }
81
82  class Print {
83      final private BufferedWriter bw;
84
85      public Print() {
86          bw = new BufferedWriter(new OutputStreamWriter(System.out));
87      }
88
89      public void print(String str) throws IOException {
90          bw.append(str);
```

```
91          }
92
93      public void printLine(String str) throws IOException {
94          print(str);
95          bw.append("\n");
96      }
97
98      public void close() throws IOException {
99          bw.close();
100     }
101 }
```

