# OpenReview forum: "CodeComplex: A Time-complexity Dataset for Multi-language Source Codes"
_ICLR.cc/2024/Conference — ICLR 2024 Conference Withdrawn Submission_

### Official Review · Reviewer_UsDM · 2023-10-24

**Soundness:** 2 fair
**Presentation:** 3 good
**Contribution:** 2 fair
**Rating:** 3
**Confidence:** 4

**Summary:**

This paper introduces the CodeComple dataset, which is the largest code dataset tailored for complexity prediction. Specifically, this dataset provides approximately 5,000 codes in both Python and Java, with a team of code specialists curating the complexity labels. Comprehensive experiments employing both traditional ML models and cutting-edge programming language models underscore the broad utility of the CodeComple dataset.

**Strengths:**

1. The presentation of this paper is clear and easy to follow, making it accessible even for readers unfamiliar with the topic.
2. The CodeComplex dataset is the largest code dataset designed specifically for the analysis of code complexity.
3. Both the dataset and the corresponding code have been provided.  However, the provided resources are not well-documented and easily accessible.
4. The paper has conducted a wide range of experiments, highlighting the broad applicability of the CodeComplex dataset.

**Weaknesses:**

## Major:

1. While this work adds value, its contributions seem to be on the incremental side. Two prior studies have already developed code complexity datasets. Beyond the addition of more Java and Python codes, this work doesn't seem to bring novel designs to the CodeComplex dataset.
2. The scope of the CodeComplex dataset appears to be limited. Ideally, a dataset focusing on code complexity should consider both time and space complexities. Yet, this particular dataset solely focuses on time complexity. It is noteworthy that some algorithms may sacrifice space complexity for prioritizing time complexity (e.g., Hash Table), or vice versa (e.g., Bitsets). Therefore, a dataset for the code complexity should consider both the space and time complexities.

3. The dataset considers only Java and Python, albeit both being among the widely used programming languages. Furthermore, the dataset's data source is confined to CodeForces. It's worth noting that there are many open-source platforms available, such as GitHub and Leetcode.
4. The documentation of this dataset raises concerns. I appreciate the authors' initiative in releasing the dataset and code on the anonymous platform. However, the anonymous GitHub repository lacks a tutorial for utilizing the dataset and essential instructions for reproducing the experiments.



## Minor:

1. In the manuscript, kindly include a reference to the Appendix. For instance, "An overview of our CodeComplex dataset can be found in Appendix A."
2. To enhance readability, please switch the positions of Figure 1 and Table 1.
3. Please remove the "For future work" at the end of the conclusion section.
4. On Page 8, please minimize the spacing, particularly below Figure 2 and above "Programming Language Understanding Models."
5. This paper mentions two previous code complexity datasets. Please incorporate both of them in Table 1.
6. In Table 3, please add the "%" symbols in 7th and 10th rows.

**Questions:**

1. As listed in Table 1, this study has intentionally curated a balanced dataset across various classes. While a balanced dataset simplifies the training of neural networks, an imbalanced distribution is often closer to real-world scenarios. Could you explain your motivation to opt for a balanced dataset?
2. The space complexity is also an important aspect of the code complexity. Why does this paper only consider the time complexity?
3. Would you consider incorporating code from diverse data sources? For example, the Leetcode website offers ground-truth time and space complexity for various algorithms, implemented in multiple program languages.

---

### Official Review · Reviewer_Vi2o · 2023-11-01

**Soundness:** 2 fair
**Presentation:** 2 fair
**Contribution:** 2 fair
**Rating:** 3
**Confidence:** 4

**Summary:**

This paper proposes a new code complexity dataset based on programming problems from Codeforces, and have human annotate the algorithm complexity of the code. The dataset is several times larger than that from previous works. The paper presents the results of several models trained on this dataset.

**Strengths:**

The new code complexity dataset is an interesting dataset, and could bring value to the community.

**Weaknesses:**

- What do you mean "OpenAI introduced GitHub Copilot, powered by Codex"? This is false statement. Github Copilot is from Microsoft, and the backend model probably is not Codex.

- Could you add experiments on the current popular LLMs' performance on the complexity datasets, like GPT-4, GPT-3.5-Turbo, StarCoder, etc. with zero-shot or few-shot? This might make the paper a lot more interesting.

- Do you really have to say ML vs DL? please don't write it like that. DL is one family in ML.

- The paper, however, lack sufficient novelty for ICLR.

**Questions:**

- Please present results on GPT-4, GPT-3.5-Turbo or something similar.
- The paper didn't mention the details of training. What goes into the model? Is it just the code part, or problem descriptions and constraints are also going into model? Without these or information on what might goes into model, it seems hard to understand the complexity of the code, even for human.

---

### Official Review · Reviewer_oyNS · 2023-11-01

**Soundness:** 2 fair
**Presentation:** 2 fair
**Contribution:** 2 fair
**Rating:** 5
**Confidence:** 3

**Summary:**

This paper presents a dataset for predicting the time complexity of coding problems. It aims to evaluate ML models' ability to understand the computational complexity of algorithms. The proposed dataset expands CoRCoD by adding more examples and more complexity categories. The paper also measures and analyzes the classification performance of classical machine learning methods with language models, including CodeBERT, GraphCodeBERT, PLBART, CodeT5, CodeT5+, and UniX-coder as baselines.

**Strengths:**

1. This work provides an extensive dataset for learning time complexity with Java and Python. This can contribute to training and improving language models of code.
2. The paper is easy to follow. The dataset creation pipeline is clearly presented.
3. The analysis with baselines is helpful for better understanding the dataset and models.

**Weaknesses:**

1. Soundness: The annotation rules take into consideration (1) problems' input dimension, (2) library usages, (3) test cases, and (4) constant input. These annotation rules are confusing: (1) considers only the largest input dimension, which is unsuitable for problems such as graph traversal; (3) fails to consider uncovered test cases.
2. The data collection and annotation process are briefly mentioned in Section 2.1 and Appendix A1 and A2. The annotation rules are unclear and, at the same time, lack demonstrations to help annotators understand the annotation process. The demographics (e.g., how many annotators are proficient programmers, consensus, etc.), the annotator training procedure (if there's one), and the annotation statistics (e.g., confidence, Kappa score, etc.) are not provided.
3. While I appreciate the effort of creating a new dataset (approximately 9 times larger than the previously largest one), this dataset seems relatively small for training the latest large language models. As in the performance section (Table 3), the performance gain is not consistently higher for language models compared to the classical methods.
3. The writing can be improved. The current version seems to be not well formatted for a high-quality paper.

**Questions:**

1. Though the baseline models in the paper are for code understanding, there are different and latest language models of code, such as CodeX, CodeGen (https://arxiv.org/abs/2305.02309), Incoder (https://arxiv.org/abs/2204.05999) with higher code prediction performance. Can they be utilized for code understanding, e.g., fine-tuning with prompting with CodeX?
2. Why do we want to eliminate dead code? Shouldn't the ability of models to "understand code" imply being able to ignore such code segments?
3. For performance in Table 3, it doesn't seem no language model performs better for most classes of complexity than others, including the classical ML method like decision tree. Does it indicate some difference between these classes, and if the datasets are actually useful for training large language models? Do you have a justification for that?

---

### Official Review · Reviewer_aZXg · 2023-11-04

**Soundness:** 3 good
**Presentation:** 2 fair
**Contribution:** 3 good
**Rating:** 5
**Confidence:** 4

**Summary:**

This paper introduces a dataset for classifying the time complexity of 4.9K Java and 4.9K Python programs from a competitive programming benchmark (Codeforces). The dataset is based on part of the Codecontests dataset. Human annotators labelled code examples with one of 7 time complexity classes (ranging from O(1) to NP-hard), using algorithmic tags for the code and a manual examination of the code and libraries that it calls. The paper fine-tunes a variety of code LLMs (CodeBERT, CodeT5, etc) on the task and reports accuracies per-model and per-time class.

**Strengths:**

S1) The benchmark is larger than previous time complexity benchmarks, both in the number of problems (~9K, compared to ~3K in the contemporaneous Tasty benchmark, Moudalya et al., and 900 in CoRCoD), the languages covered (two, compared to one in CoRCoD), and the number of time complexity classes (7, compared to 5 in CoRCoD).

S2) The benchmark appears to be fairly carefully constructed (but some more details on this would be helpful, see below): introducing some data augmentation approaches and dead code elimination, and having human annotators label the code and verify each others' labels.

S3) Some aspects of the experimentation were thorough, involving a large number of strong code encoding models.

**Weaknesses:**

The contribution overall felt a bit thin to me.

W1) It would help to do some experiments to demonstrate the benefit of this larger dataset, by e.g. showing performance by amount of training data (show sample complexity curves). I think this is especially important given that some fraction of the dataset seems to have been generated using data augmentation (details of this, and its size) were a bit unclear to me.

W2) While there are lots of results from various models, there were a limited number of clear takeaways. To have more solid comparisons between the models, I think that the paper should tune their learning rates on a validation set, rather than using the same LR for all models. The accuracy-by-length is an interesting step toward a fine-grained analysis (as are the per-class analysis, but see below), but would be further improved by looking at other attributes of the code, e.g. the per-algorithm accuracies.

W3) The paper should do a bit more to differentiate itself from past work, especially TASTY (Moudgalya 2023). While I appreciate that TASTY came out not so long before the ICLR submission deadline (~4 months), it does seem similar enough to this benchmark that it would be helpful to e.g. compare findings between models on the two benchmarks, or at least say more about the differences in the paper.

W4) It would help to improve the motivation for the work -- what would the value be of a model that can predict time complexity? e.g. some experiments showing that reranking with such a model can improve the time complexity of generated code, or be used in the sort of pedagogical applications mentioned in the intro.

W5) The writing could be improved in clarity, see below.

**Questions:**

Q1) I had a few questions about the annotation process:

Q1a) "Consider the input size as a parameter to determine the time complexity of a code. We measure the complexity by the largest parameter among them.". I'm unclear what this second sentence means: does it refer to the largest possible input to the code?

Q1b) "When a problem provides a fixed constant as input, we classify the case as having a constant time complexity.". Could more detail be given about this assumption: (a) I'm unclear on when a problem would need to have a fixed constant as an input, and (b) it seems that some problems that don't take any inputs might have non-constant time complexity (i.e. program that prints the first 100K prime numbers might take no inputs, but have high time complexity).

Q1c) How did the annotators resolve the ambiguous scenarios mentioned at the bottom of page 3? e.g. when the problem itself has a higher time complexity than the problem+input constraints, which is used for the time complexity?

Q2) How many instances are in the augmented dataset, versus in augmented? Does the "random" split in the experimental setup take augmentation into account (no overlap in augmented code between train and test)?

Q3) I'm unclear on how the scores in Table 3 were obtained. Is the prediction task here a 7-way prediction, but divided up by the ground-truth class?

Other suggestions
- It would help to make a pass over the paper for clarity and grammar. e.g. the final section ends mid-sentence.